# Serum Lactate Dehydrogenase Level One Week after Admission Is the Strongest Predictor of Prognosis of COVID-19: A Large Observational Study Using the COVID-19 Registry Japan

**DOI:** 10.3390/v15030671

**Published:** 2023-03-02

**Authors:** Sho Nakakubo, Yoko Unoki, Koji Kitajima, Mari Terada, Hiroyuki Gatanaga, Norio Ohmagari, Isao Yokota, Satoshi Konno

**Affiliations:** 1Department of Respiratory Medicine, Faculty of Medicine, Hokkaido University, Sapporo 060-8638, Japan; 2Department of Biostatistics, Graduate School of Medicine, Hokkaido University, Sapporo 060-8638, Japan; 3Centre for Clinical Sciences, National Centre for Global Health and Medicine, Tokyo 162-8655, Japan; 4Disease Control and Prevention Centre, National Centre for Global Health and Medicine, Tokyo 162-8655, Japan; 5AIDS Clinical Centre, National Centre for Global Health and Medicine, Tokyo 162-8655, Japan; 6AMR Clinical Reference Centre, National Centre for Global Health and Medicine, Tokyo 162-8655, Japan; 7Institute for Vaccine Research and Development, Hokkaido University, Sapporo 060-8638, Japan

**Keywords:** COVID-19, COVID-19 registry Japan, lactate dehydrogenase, SARS-CoV-2

## Abstract

Clinical features of COVID-19 are diverse, and a useful tool for predicting clinical outcomes based on clinical characteristics of COVID-19 is needed. This study examined the laboratory values and trends that influence mortality in hospitalised COVID-19 patients. Data on hospitalised patients enrolled in a registry study in Japan (COVID-19 Registry Japan) were obtained. Patients with records on basic information, outcomes, and laboratory data on the day of admission (day 1) and day 8 were included. In-hospital mortality was set as the outcome, and associated factors were identified by multivariate analysis using the stepwise method. A total of 8860 hospitalised patients were included. The group with lactate dehydrogenase (LDH) levels >222 IU/L on day 8 had a higher mortality rate compared to the group with LDH levels ≤222 IU/L. Similar results were observed in subgroups formed by age, body mass index (BMI), underlying disease, and mutation type, except for those aged <50 years. When age, sex, BMI, underlying disease, and laboratory values on days 1 and 8 were tested for factors strongly associated with in-hospital mortality, LDH on day 8 was most strongly associated with mortality. LDH level on day 8 was the strongest predictor of in-hospital mortality in hospitalised COVID-19 patients, indicating its potential usefulness in post-treatment decision-making in severe COVID-19 cases.

## 1. Introduction

Coronavirus disease 2019 (COVID-19), an infectious disease caused by severe acute respiratory syndrome coronavirus 2 (SARS-CoV-2), has affected a large number of individuals and resulted in over six million deaths worldwide [1]. Proven-effective treatments have been put into practice, and vaccines have been popularised [2,3]. In addition, as a result of the transition in the virulence of the SARS-CoV-2 variant, the rate of severe illness and mortality has decreased compared to the beginning of the epidemic [4,5]. However, the fact remains that some patients develop severe disease with long duration with sometimes fatal outcomes. It is continually important to find practical clinical prediction tools that consider the risk of poor outcomes for patients and allocate healthcare resources appropriately.

Several factors predispose patients to severe disease and death from COVID-19, including advanced age, obesity, and underlying diseases [6]. Results of a large registry study suggested that scoring risk factors from patient backgrounds can predict prognosis in COVID-19 patients [7]. In addition, patients with severe disease are more likely to have decreased lymphocyte counts and elevated levels of enzymes, inflammatory markers, and D-dimer [8,9]. Some previous studies have also focused on novel biomarkers, such as soluble urokinase receptors and C-reactive protein isoforms [10,11]. Clinical prediction tools combining these easily measurable but nonspecific laboratory markers have also been proposed [12,13]. Although they are useful in estimating the risk of resistance to treatment and mortality in COVID-19 patients, most are evaluated only during the initial visit or on admission. The clinical course of COVID-19 is varied and long. Hence, evaluation of laboratory values at the initial point may fail to identify patients with potentially poor prognoses or overestimate the risk of patients who respond rapidly to initial treatment. Focusing on peak laboratory values has been reported to improve the accuracy of predicting death [14], but laboratory value trends vary from patient to patient and are difficult to apply in actual clinical scenarios. How changes in laboratory values over a short period of time affect the prognosis of patients with COVID-19 is not well understood.

Serum lactate dehydrogenase (LDH) levels are often elevated in patients with severe COVID-19, reflecting extensive pulmonary damage [15]. Elevated serum LDH at initial presentation has been shown to strongly influence progression to respiratory failure and death; therefore, LDH level is regarded as a useful indicator to initiate close monitoring to prevent poor outcomes in patients with COVID-19 [16,17,18]. Although critically ill patients and those with fatal courses are known to have higher serum LDH levels over a long clinical course [19], the prognostic impact of LDH level trends remains unknown.

In Japan, a large-scale registry study (COVID-19 REGISTRY JAPAN, COVIREGI-JP) is underway to collect clinical information on COVID-19 patients from multiple centres [20]. Using this registry data, our study aimed to validate two-time point laboratory values affecting mortality in hospitalised patients with COVID-19 and to find a universal and simple prognostic marker independent of broad patient status at admission.

## 2. Materials and Methods

### 2.1. Study Design and Participants

This was an observational study, and the participants were enrolled from medical institutions participating in COVIREGI-JP. Criteria for enrolment in COVIREGI-JP were: (1) positive SARS-CoV-2 test and (2) hospitalisation and treatment at a registered medical institution. Study data were collected and managed using Research Electronic Data Capture (REDCap), a secure, web-based data capture application hosted at the Joint Centre for Researchers, Associates, and Clinicians (JCRAC) Data Centre of the National Centre for Global Health and Medicine. We excluded patients who were transferred from other hospitals since their progress from the start of the treatment was unknown.

### 2.2. Study Period

Hospitalised patients enrolled between 20 March 2020 and 30 June 2021 were included in the study. Owing to the revision of COVIREGI-JP case registration form, the period after July 2022 was excluded since registration of post-hospitalisation laboratory data was no longer required after that date.

### 2.3. Data Collection

Demographic characteristics (age, sex) and clinical data (body mass index [BMI], comorbidities, laboratory data, and in-hospital death) were collected. The date of admission was defined as day 1, according to the data entry rules of COVIREGI-JP. In accordance with the COVIREGI-JP data entry regulations, laboratory data collected on days 0–3 were pooled and registered as day 1, while those collected on days 7–9 were pooled and registered as day 8. Day 0 laboratory data refers to the data collected on the day before admission for a patient transferred from another hospital. Blood laboratory investigations analysed in this study included white blood cell (WBC) count, lymphocyte fraction, platelet count, aspartate aminotransferase (AST), alanine aminotransferase (ALT), LDH, creatine kinase (CK), C-reactive protein (CRP), and creatinine. Other laboratory investigations, such as ferritin, D-dimer, and procalcitonin, are factors that may influence the prognosis of COVID-19 [9]. However, only a few cases were entered with these data and were therefore excluded from this study. Although the reference values for laboratory data differ depending on the facility and measuring instruments, the reference value set at Hokkaido University Hospital was adopted in this study. A value of 222 IU/L was used as the upper limit for LDH. Cases in which this information was registered without deficiencies were included in the analysis. Additional data on respiratory status at the time of admission were collected.

### 2.4. Clinical Outcomes

Death during hospitalisation was set as the clinical outcome.

### 2.5. Statistical Analysis

We defined LDH levels as low range (≤222 IU/L) and high range (>222 IU/L). The eligible patients were classified into four groups based on their LDH results on day 1 and day 8. The groups were as follows: (1) high–high (H–H), (2) high–low (H–L), (3) low–high (L–H), and (4) low–low (L–L). For example, H–L group described patients who had an LDH level trend within the elevated range (>222 IU/L) on day 1 and then the low range (≤222 IU/L) on day 8. Patient characteristics were compared between the groups. Continuous and categorical variables were presented as medians and interquartile ranges (IQRs) and counts and proportions, respectively. Continuous and categorical variables were analysed using the one-way analysis of variance and Pearson’s chi-square, respectively.

We compared the in-hospital mortality between the groups using descriptive statistics. Additionally, we conducted subgroup analyses stratified by sex, BMI (<18.5, 18.5–24.9, 25–29.9, >30 kg/m^2^), age (<50, 50–59, 60–69, 70–79, >80 years), comorbidity (cardiovascular diseases, respiratory diseases, liver diseases, renal diseases, diabetes, neoplasms, and cerebrovascular diseases), and variant strains of SARS-CoV-2 (alpha and delta). The alpha cohort comprised patients diagnosed between 17 March 2020 and 31 March 2021, while the delta cohort consisted of patients diagnosed between 1 April and 30 June 2021. Multivariate binary logistic models were then produced using a forward–backward stepwise approach, using the Akaike Information Criterion (AIC). The models were retained with the following variables (age, sex, BMI, and comorbidities). To identify the prognostic impact of LDH level trends, we analysed the data using three different timing-tested laboratory data. Each model contained laboratory data obtained on days 1, 8, and both. Receiver operating characteristic (ROC) curves were then generated, and the area under the receiver curves (AUCs) were calculated separately for the models and LDH level on day 8 alone to determine discrimination. Additionally, we calculated the sensitivity, specificity, positive predictive value (PPV), and negative predictive value (NPV) at various cut-off points of LDH level on day 8. 

All cases with missing data for any of the selected covariates, except respiratory status on admission, were excluded from the analysis. R software (version 3.5.1; R Foundation for Statistical Computing, Vienna, Austria) was used for all statistical analyses. All confidence intervals (CIs) were set at 95%.

## 3. Results

A total of 47,355 hospitalised patients were registered in the COVIREGI-JP between 20 March 2020 and 30 June 2022. Patients transferred from other hospitals were excluded. Clinical characteristics and laboratory values of days 1 and 8 were described without excess or deficiency in 8860 patients. The clinical data of these 8860 patients were included in the analysis (Figure 1). 

We examined the clinical characteristics of the patients in total and of the four groups classified on the basis of LDH levels. The median age of total patients (*n* = 8860) was 65 years, the median BMI was 23.8 kg/m^2^, and 39.4% were females. Diabetes mellitus was the most common underlying medical condition (21.8%), followed by chronic respiratory diseases (10.2%). At the time of admission, 81.8% of patients did not require oxygen, 17.8% needed oxygen, invasive mechanical ventilation was performed in 32 patients (0.4%), and non-invasive positive pressure ventilation was used in only three patients (<0.1%). The results of laboratory investigations are shown in Table 1. The median LDH level was 246 IU/L on day 1 and 238 IU/L on day 8. A total of 475 (5.4%) patients died during hospitalisation. When the patients were divided into four groups (H–H, H–L, L–H, and L–L groups) according to the LDH level on days 1 and 8, differences were observed among the four groups in terms of median age, percentage of men and women, median BMI, percentage of underlying disease, and percentage severity of respiratory status at admission. Median age, BMI, and proportion of females were higher in the H–H and L–H groups compared to the H–L and L–L groups. The H–H group had the highest percentage of patients requiring oxygen or ventilators on admission, followed by the H–L group. Considerable differences were observed in the median values of laboratory parameters (WBC count, lymphocyte fraction, platelet count, AST, ALT, LDH, CK, CRP, and creatinine) measured between the four groups on days 1 and 8 (Table 1). 

Next, differences in mortality rates among the four groups based on LDH values were calculated. The group with high LDH levels (>222 IU/L) on day 1 had a significantly higher mortality rate compared to the group with low LDH levels (≤222 IU/L) on day 1 (7.2 vs. 2.5%). Among patients with high LDH levels on day 1, a large difference in mortality rate was observed between the H–H and H–L groups (9.4 vs. 0.7%) (Table 1, Figure 2). Contrarily, among patients who did not have high LDH levels on day 1, the L–H group had a higher mortality rate than the L–L group (6.8 vs. 0.5%) (Table 1, Figure 2). Subgroups were formed by sex, age, BMI, underlying diseases, and dominant strains of SARS-CoV-2, and mortality rates of the four groups were compared based on LDH values. In all subgroups, the mortality rates of H–H and L–H groups were higher compared to H–L and L–L groups, except for the population aged <50 years (Figure 2). 

Multivariate analysis using the stepwise method was performed to examine how each factor affected in-hospital mortality and to calculate the best prediction model. First, patient characteristics (age, sex, BMI, comorbidities) were set as fixed variables, and other variables were selected from day 1 laboratory values using the stepwise method. The results showed that lymphocyte fraction, platelet count, creatinine, CRP, and CK were independent prognostic factors, while LDH was not chosen as the optimal variable (appendix, Appendix A). In contrast, when the prediction model was validated by employing laboratory values from both days 1 and 8, LDH levels on day 8 were selected as the variable with the strongest effect on death among all the variables (adjusted OR [95% CI]: 1.006 [1.005–1.007], chi-squared value 250.747, *p* < 0.001) (Table 2; appendix, Appendix A). Other elements that were selected as the best model for predicting mortality were lymphocyte fraction on day 8, CRP on day 8, platelet count on day 8, creatinine on day 1, CK on day 8, and CRP on day 1 (adjusted OR [95% CI]: 0.893 [0.875–0.911], 1.018 [1.011–1.026], 0.998 [0.997–0.999], 1.098 [1.017–1.186], 1.000 [0.999–1.000], and 1.001 [1.000–1.002], respectively) (Table 2; appendix, Appendix A). 

Finally, we drew ROC curves for factors predicting in-hospital mortality and tested the utility of LDH on day 8 in building the prediction models. When age, sex, BMI, and underlying diseases were set as variables in the baseline prediction model, the AUC for the ROC curve was 0.84. Contrarily, the AUC of the ROC curve for LDH on day 8 alone was 0.88 (Figure 3A). When the cut-off value (IU/L) for LDH was set at 222, the sensitivity was 95.8% (95% CI: 93.6–97.4%), specificity was 43.3% (95% CI: 42.3–44.4%), PPV was 8.7% (95% CI: 8.0–9.5%), and NPV was 99.5% (95% CI: 99.2–99.7%). When set at 444, the sensitivity, specificity, PPV, and NPV were 49.7% (95% CI: 45.1–54.3%), 95.2% (95% CI: 94.7–95.6%), 36.8% (95% CI: 33.1–40.7%), and 97.1% (95% CI: 96.7–97.4%), respectively. Detailed results of the cut-off values of LDH, sensitivity, and specificity for in-hospital death are shown in the appendix, Appendix A. We then added LDH on day 8 as a variable to the baseline model and obtained an AUC of 0.93. When lymphocyte fraction, the second most influential factor on mortality on day 8 blood counts in the multivariate analysis was included in the baseline model, the AUC was 0.92. The optimal day 8 blood laboratory values to be added to the baseline model were selected using a stepwise method: LDH, lymphocyte count, CRP, platelet count, CK, and creatinine. The AUC of the ROC curve drawn based on the model was 0.95. (Figure 3B). The results of the multivariate analysis of the predictive model corresponding to the ROC curves in Figure 3B are shown in the appendix Appendix A.

## 4. Discussion

This study examined the predictors of in-hospital mortality in hospitalised COVID-19 patients using data from a nationwide registry study in Japan. A large difference was observed in patient mortality between high and low serum LDH values measured on day 8 of hospitalisation, with similar results in most subgroups. Multivariate analysis revealed that LDH levels on day 8 were the most influential and independent prognostic factor predicting poor outcomes for in-hospital mortality in COVID-19 patients. The ROC curve for predicting death with day 8 LDH values showed an AUC of 0.88, with high NPV at a cut-off value of 222 IU/L and high PPV at a cut-off value of 444 IU/L.

It is challenging to establish an accurate and useful mortality prediction model for COVID-19. While some patients with COVID-19 become severely ill within approximately eight days of onset and critical or fatal by approximately 16 days, most only have mild symptoms at the time of disease onset [15]. Even when pneumonia sets in and the disease becomes severe, only a small percentage of cases are fatal, making it difficult to estimate the risk of dying early in the course of the disease or at the time of hospitalisation. Previous studies have shown that evaluation of LDH and other laboratory values at the time of admission can predict subsequent severity of illness and death among patients [16,17,21]. Contrarily, the results of our study indicated that serum LDH levels at the time of admission had a limited impact on predicting patient mortality. In Japan, medical resources vary from region to region, and the threshold for patient hospitalisation and time required for admission also differs depending on the prevalence of COVID-19. Hospitalised COVID-19 patients were a heterogeneous population in terms of the time of onset and risk of severe illness or death. The predictive model, which was calculated by combining a large amount of clinical information, showed high predictive accuracy. However, it is a cumbersome process and requires a special method for calculation [22,23].

In this study, we included laboratory data from approximately eight days after admission. Our findings showed that mortality prediction became more accurate by including data from day 8 than using data from the day of admission and that the effect of LDH, in particular, had a significant impact on mortality. Considering the long course of severe to critical COVID-19, prognosis prediction using day 8 laboratory data would be pragmatic. A lower cut-off (222 IU/L) for day 8 LDH value may be useful in determining discharge or early completion of treatment because the PPV for death is comparatively low for patients with low LDH levels. In contrast, a higher cut-off (LDH 444 IU/L) increased the PPV for death. Thus, this factor can be considered when making an informed decision regarding reconsideration of the treatment plan in patients who have a poor response to initial treatment for COVID-19; any treatment changes should be made after obtaining informed consent from the patient.

A noteworthy aspect of this study was the finding of the usefulness of test items that can be measured universally, regardless of region or facility. In addition, the ROC curve for predicting death based on LDH level on day 8 showed a high AUC without considering the baseline characteristics and severity of COVID-19 at the time of admission. We also showed that more accurate prognostic models could be constructed by incorporating age and underlying disease, in addition to serum LDH levels on day 8. Nevertheless, the process of checking a single LDH level after admission is simple and easy to perform in actual clinical practice. It could provide universal benefits independent of medical resources and social environment.

LDH is a nonspecific enzyme which is elevated in malignancies, liver diseases, interstitial pneumonia, and other infections [24,25,26,27]. Hence, some of the enrolled patients might have had elevated LDH from the outset, reflecting an underlying disease; and deaths in the group with elevated LDH from admission might have included a small number of deaths due to the underlying condition. However, the fact that LDH on day 8 had a stronger effect on COVID-19 mortality than any other factor in the multivariate analysis suggests that elevated LDH levels primarily reflect organ injury caused by COVID-19. COVID-19 causes not only extensive pulmonary damage but also myocardial and microvascular damage, both of which can result in LDH deviation [15,28,29]. Both conditions are complicated in the severe phase of the disease and carry a high mortality rate. A significant risk of death with high serum LDH on day 8 may reflect the presence of these late complications.

Our study did not include treatment for COVID-19 in the analysis because the exact timing of initiation of treatment or doses for medications such as steroids was not recorded in several cases registered in the COVIREGI-JP. Steroids and immunosuppressive drugs have been used in a substantial percentage of patients in clinical practice. This may reflect that CRP levels on day 8 were not significantly associated with death in our study, although inflammatory reactions have been predictive of death in previous reports [9,30]. It is possible that treatment for COVID-19 was also a significant confounding factor for LDH. However, in light of the results of this study, unlike the inflammatory response markers, LDH may be a less modifiable factor in the treatment of COVID-19. This suggests that the LDH level is probably the laboratory value that reflects the true response of COVID-19 pneumonia to treatment.

We found that the peripheral blood lymphocyte fraction on day 8 was the second strongest predictor of mortality among the laboratory values after LDH. In severe cases of COVID-19, viral attachment to lymphocytes, damage by inflammatory cytokines, migration, and aggregation to lung tissue are the causes of persistent decline in peripheral blood lymphocytes [31]. Decreased peripheral blood lymphocyte counts in COVID-19 patients have been widely reported as a risk factor for severe disease [9,13,31]. Our study showed that the effect of COVID-19 on lymphocytes was not attenuated on day 8, indicating that lymphopenia could be a major factor in predicting poor prognosis. Using the ratio of LDH to lymphocyte count [32], as suggested by a previous study, devising a prediction model that combines multiple laboratory values with strong influences may prove to be useful. However, in clinical practice, where versatility and simplicity are required, there is no substitute for the utility of a single LDH level measurement at one-week post-hospitalisation.

This study had several limitations. First, its retrospective, observational study design. Second, the data registration of COVIREGI-JP is dependent on the voluntary effort of participating centres, which may have led to a selection bias in the enrolled cases. Additionally, as the clinical outcome was defined as death during hospitalisation, long-term prognosis, such as death after transfer, was not considered. Some factors that could affect outcomes, such as mutant strain and vaccination history, were not evaluated. Some of the established predictors of COVID-19 severity, such as serum procalcitonin, ferritin, and D-dimer, were not analysed due to the limited number of patients for whom data on these items were entered. Furthermore, as indicated in the Materials and Methods section, there were discrepancies in the timing of the measurement of the laboratory values collected in this study. In addition, laboratory data other than day 1 (day 0–3) and day 8 (day 7–9) have not been validated; therefore, the possibility that there were other laboratory values and their timing that were optimal for prognostic prediction cannot be ruled out. Last, in Japan, it is possible that a substantial number of severe disease cases that did not receive ventilators or other treatment were included. Therefore, the impact of LDH on mortality may have been overestimated.

## 5. Conclusions

We found that LDH levels on day 8 were the strongest predictor of in-hospital mortality in hospitalised COVID-19 patients. Easily measured LDH after initial treatment of COVID-19 is suggested to be of great help in medical care planning.

## Figures and Tables

**Figure 1 viruses-15-00671-f001:**
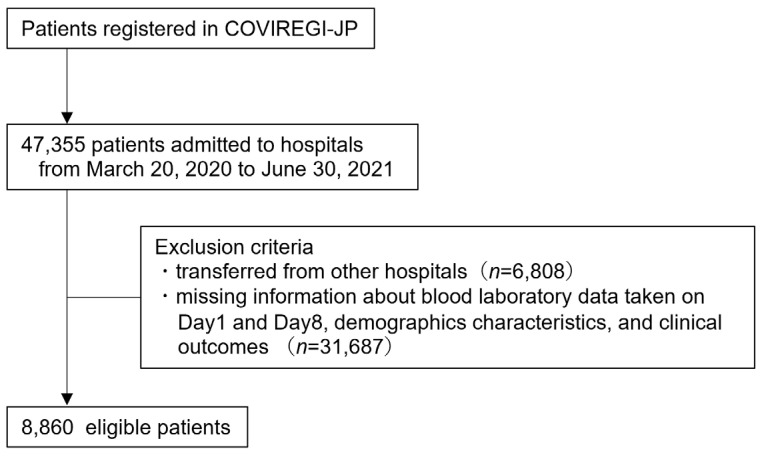
Patient selection flowchart. Abbreviations: COVIREGI-JP, COVID-19 REGISTRY JAPAN.

**Figure 2 viruses-15-00671-f002:**
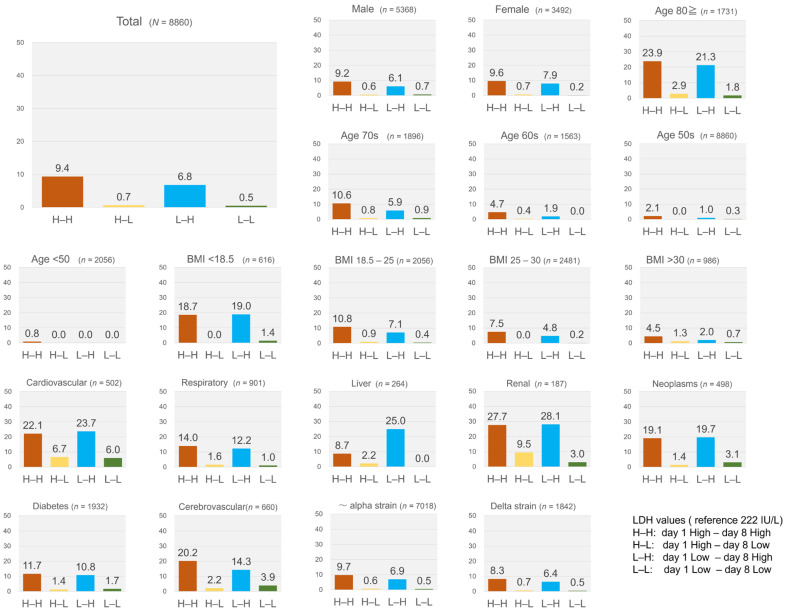
In-hospital mortality rates of the four groups classified based on LDH values. LDH level of 222 IU/ or less was defined as ‘low’, and levels above 222 IU/L were defined as ‘high’. H–H, day 1 high, day 8 high; H–L, day 1 high, day 8 low; L–H, day 1 low, day 8 high; L–L, day 1 low, day 8 low. Abbreviations: LDH, lactate dehydrogenase.

**Figure 3 viruses-15-00671-f003:**
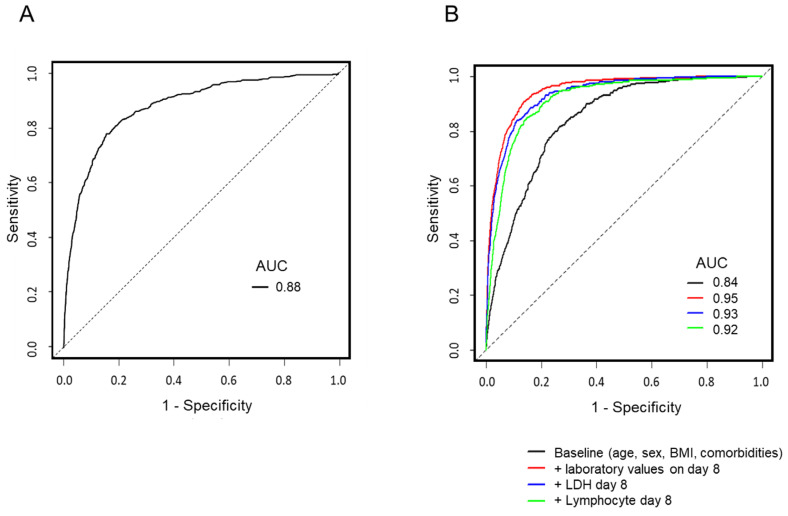
ROC curves and AUCs for models predicting in-hospital mortality. (**A**) ROC curve for LDH on day 8. (**B**) ROC curves for predicting models derived from multivariate logistic regression analysis. The results of the multivariate analysis of each model are in appendix Appendix A. Abbreviations: AUC, area under curve; LDH, lactate dehydrogenase; ROC, receiver operating characteristic.

**Table 1 viruses-15-00671-t001:** Patient characteristics and laboratory values in the groups classified by the level of serum LDH on day 1 * and 8.

	Total	H–H	H–L	L–H	L–L	*p*-Value
	(*n* = 8860)	(*n* = 4044)	(*n* = 1365)	(*n* = 1115)	(*n* = 2336)	
Patient characteristics						
Age	65.00 [51.00, 77.00]	69.00 [55.00, 78.00]	61.00 [50.00, 73.00]	68.00 [54.00, 79.00]	56.00 [39.00, 73.00]	<0.001
Female sex, *n* (%)	3492 (39.4)	1406 (34.8)	566 (41.5)	429 (38.5)	1091 (46.7)	<0.001
BMI	23.88 [21.35, 26.84]	24.51 [21.91, 27.70]	23.98 [21.70, 26.84]	24.00 [21.60, 26.56]	22.84 [20.48, 25.38]	<0.001
Comorbidities, *n* (%)						
Cardiovascular diseases	502 (5.7)	298 (7.4)	45 (3.3)	76 (6.8)	83 (3.6)	<0.001
Respiratory diseases	901 (10.2)	485 (12.0)	126 (9.2)	98 (8.8)	192 (8.2)	<0.001
Liver diseases	264 (3.0)	150 (3.7)	46 (3.4)	28 (2.5)	40 (1.7)	<0.001
Renal diseases	187 (2.1)	101 (2.5)	21 (1.5)	32 (2.9)	33 (1.4)	0.003
Neoplasms	498 (5.6)	257 (6.4)	69 (5.1)	76 (6.8)	96 (4.1)	<0.001
Diabetes mellitus	1932 (21.8)	1062 (26.3)	289 (21.2)	222 (19.9)	359 (15.4)	<0.001
Cerebrovascular diseases	660 (7.4)	337 (8.3)	91 (6.7)	105 (9.4)	127 (5.4)	<0.001
Respiratory status on admission, *n* (%)						
Room air	7217 (81.8)	2790 (69.3)	1146 (84.3)	1026 (92.7)	2255 (96.9)	<0.001
Oxygen therapy	1566 (17.8)	1200 (29.8)	214 (15.7)	80 (7.2)	72 (3.1)	
Non-invasive mechanical ventilation	3 (0.0)	3 (0.1)	0 (0.0)	0 (0.0)	0 (0.0)	
Invasive mechanical ventilation	32 (0.4)	31 (0.8)	0 (0.0)	1 (0.1)	0 (0.0)	
Missing	42 (0.5)	20 (0.5)	5 (0.3)	8 (0.7)	9 (0.4)	
Clinical outcome, *n* (%)						
In-hospital death	475 (5.4)	379 (9.4)	9 (0.7)	76 (6.8)	11 (0.5)	<0.001
Laboratory values						
WBC (×10^3^/µL)						
day1	5.10 [4.00, 6.59]	5.31 [4.13, 7.00]	5.31 [4.20, 6.80]	4.80 [3.90, 5.98]	4.70 [3.70, 5.90]	<0.001
day8	6.70 [5.00, 9.41]	8.00 [5.90, 11.05]	6.60 [5.11, 8.77]	6.30 [4.60, 9.05]	5.38 [4.30, 7.00]	<0.001
lymphocyte cell (%)						
day1	20.20 [13.70, 27.50]	17.20 [11.30, 24.00]	20.10 [14.10, 27.10]	21.60 [16.00, 28.20]	25.00 [18.00, 32.40]	<0.001
day8	20.00 [11.70, 28.60]	14.80 [8.20, 23.00]	24.00 [16.50, 30.30]	17.40 [9.65, 25.90]	27.55 [19.90, 34.80]	<0.001
Platelet count (×10^3^/µL)						
day1	167.00 [119.00, 214.00]	160.00 [116.00, 206.00]	180.00 [130.00, 232.00]	159.00 [114.00, 199.00]	175.00 [122.00, 220.00]	<0.001
day8	236.00 [152.00, 321.00]	251.00 [159.00, 343.00]	291.00 [190.00, 372.00]	196.00 [130.00, 260.50]	215.00 [140.75, 278.00]	<0.001
AST (U/L)						
day1	32.00 [23.00, 46.00]	42.00 [31.00, 60.00]	33.00 [25.00, 47.00]	26.00 [22.00, 33.00]	23.00 [19.00, 28.00]	<0.001
day8	27.00 [20.00, 40.00]	33.00 [23.00, 50.00]	24.00 [18.00, 32.00]	32.00 [24.00, 45.00]	21.00 [17.00, 27.00]	<0.001
ALT (U/L)						
day1	25.00 [16.00, 41.00]	30.00 [19.00, 49.00]	27.00 [18.00, 46.00]	21.00 [15.00, 32.00]	18.00 [13.00, 28.00]	<0.001
day8	33.00 [19.00, 59.00]	41.00 [24.00, 76.00]	34.00 [20.00, 61.00]	31.00 [20.00, 54.50]	22.00 [14.00, 37.00]	<0.001
LDH (U/L)						
day1	246.00 [198.00, 326.00]	320.00 [265.00, 413.00]	265.00 [240.00, 309.00]	199.00 [183.00, 211.00]	182.00 [162.00, 201.00]	<0.001
day8	238.00 [192.00, 308.00]	299.50 [257.00, 376.00]	196.00 [180.00, 210.00]	269.00 [240.50, 320.00]	175.00 [153.00, 196.00]	<0.001
CK (U/L)						
day1	92.00 [59.00, 159.00]	122.50 [74.00, 248.25]	84.00 [54.00, 136.00]	85.00 [58.00, 124.00]	69.00 [49.00, 100.00]	<0.001
day8	38.00 [25.00, 59.00]	40.00 [25.00, 71.00]	30.00 [21.00, 48.00]	44.00 [29.00, 73.00]	36.00 [25.00, 51.25]	<0.001
CRP (mg/dL)						
day1	2.81 [0.70, 7.13]	5.58 [2.42, 10.42]	3.91 [1.25, 7.70]	1.45 [0.53, 3.37]	0.59 [0.19, 1.96]	<0.001
day8	1.07 [0.30, 3.51]	1.63 [0.54, 4.88]	0.43 [0.17, 1.13]	3.42 [1.29, 6.74]	0.41 [0.10, 1.50]	<0.001
Creatinine (mg/dL)						
day1	0.84 [0.68, 1.02]	0.88 [0.72, 1.10]	0.82 [0.66, 0.97]	0.87 [0.71, 1.04]	0.78 [0.63, 0.93]	<0.001
day8	0.77 [0.63, 0.92]	0.77 [0.64, 0.94]	0.77 [0.65, 0.91]	0.78 [0.65, 0.95]	0.75 [0.62, 0.89]	<0.001

Data were presented as median [IQR] of patients unless otherwise indicated. * Day 1 refers to the day the patient was admitted to the hospital. The patients were classified into four groups according to their LDH test results· LDH level 222 IU/ or less was defined as ‘low’, and LDH level over 222 IU/L was defined as ‘high’· H–H, day 1 high, day 8 high; H–L, day 1 high, day 8 low; L–H, day 1 low, day 8 high; L–L, day 1 low, day 8 low. Abbreviations; ALT, alanine aminotransferase; AST, aspartate aminotransferase; BMI, body mass index; CK, creatine kinase; CRP, C-reactive protein; LDH, lactate dehydrogenase; WBC, white blood cell.

**Table 2 viruses-15-00671-t002:** Factors associated with in-hospital mortality selected for multivariate analysis using the stepwise method (including laboratory values on day 1 * and 8).

Variables	Estimate	Standard Error	χ2	*p*-Value
Sex	−0.2275	0.1298	3.070	0.09
Age	0.0818	0.0067	149.500	<0.001
BMI	−0.0307	0.0149	4.252	0.05
Cardiovascular diseases	0.4444	0.1628	7.458	0.005
Respiratory diseases	0.4277	0.1600	7.145	0.005
Liver diseases	0.5002	0.3065	2.663	0.11
Renal diseases	0.4008	0.3088	1.685	0.19
Neoplasms	0.5944	0.1786	11.076	<0.001
Diabetes mellitus	0.3777	0.1296	8.491	0.004
Cerebrovascular diseases	0.3291	0.1577	4.356	0.04
LDH_day8	0.0062	0.0004	249.798	<0.001
lymphocyte_day8	−0.1131	0.0100	126.630	<0.001
CRP_day8	0.0185	0.0035	27.301	<0.001
Platelets_day8	−0.0018	0.0005	11.042	<0.001
Creatinine_day1	0.0939	0.0394	5.693	0.02
CK_day8	−0.0003	0.0002	4.866	0.03
CRP_day1	0.0006	0.0005	1.628	0.20

* Day 1 refers to the day the patient was admitted to the hospital. Abbreviations: BMI, body mass index; CI, confidential interval; CK, creatine kinase; CRP, C-reactive protein; LDH, lactate dehydrogenase; OR, odds ratio; WBC, white blood cell.

## Data Availability

The datasets generated and analysed in this study are not publicly available due to the COVIREGI-JP regulations. Portions of the data will be made available upon reasonable request to the corresponding author.

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
