# Peer review of "Serum Lactate Dehydrogenase Level One Week after Admission Is the Strongest Predictor of Prognosis of COVID-19: A Large Observational Study Using the COVID-19 Registry Japan"

_viruses, 2023, doi:10.3390/v15030671_

Round 1
Reviewer 1 Report
Reviewer report:
Ethical issues
The study has a major ethical problem. The authors discuss about LDH values as being useful in determining whether treatment withdrawal and palliative care are warranted in some patients due to poor response to initial treatment for COVID-19 (rows 272-274).
Abstract
The conclusion from the abstract was LDH level on day 8 was the strongest predictor of in-hospital mortality in hospitalised COVID-19 patients, indicating its potential usefulness in post-treatment decision-making in severe COVID-19 cases. However, the conclusion from the manuscript says nothing about severe cases (We found that LDH levels on day 8 were the strongest predictor of in-hospital mortality 330 in hospitalized COVID-19 patients) since not every hospitalised patients has a severe disease. Moreover, in the methodology the severe form of COVID-19 is not defined and most patients included in this study had non-severe disease (81.8%, Table 1, variable Room air), at least at hospital admission as nothing else is mentioned about clinical form of the disease after this table and it is not included in the analysis of mortality which is also a major issue.
Introduction section
The aim of the study is not well defined, in the abstract, but also in the manuscript.
Material and methods section
This section has a major issue for the analysis. The eligible patients were classified into four groups based on their LDH results on day 1 and day 8 (rows 107-108) and the authors mention different days for laboratory data collection Laboratory data on admission and day 8 included the results obtained on days 0–3 and 7–9, respectively (rows 91-92).
Results section
Other major issues could be found in Table 1:
§ from the total of 8860 patients, 475 patients died (5.4% in hospital mortality rate) but only 35 patients were mechanically ventilated, assuming these patients were hospitalised in the ICU; what happened with the rest of the patients that died? Was that on a normal ward? (authors need to explain);
§ those are only descriptive values, a statistical test was not performed; the p values are missing; the authors could use a Kruskal Wallis test (?);
In the results section Considerable differences were observed in the median values of laboratory parameters (WBC count, lymphocyte fraction, platelet count, AST, ALT, LDH, CK, CRP, and creatinine) measured between the four groups on days 1 and 8 (Table 1) (rows 162-164). A statistical test was not used to support these results.

Author Response
Reviewer 1
Ethical issues
The study has a major ethical problem. The authors discuss about LDH values as being useful in determining whether treatment withdrawal and palliative care are warranted in some patients due to poor response to initial treatment for COVID-19 (rows 272-274).
RESPONSE:
Thank you very much for your constructive feedback to further improve our paper.
Based on our experience in treating patients with severe COVID-19 disease, we have searched for useful criteria to determine the appropriate time to discontinue treatment or transition to palliation. However, as you have pointed out, the description may propose an ethical conflict. The sentence has been revised to indicate that it can contribute to making an informed decision regarding reconsidering treatment plans, and that informed consent should be obtained from patients.
Lines 284–286:
“Thus, this factor can be considered when making an informed decision regarding reconsideration of the treatment plan in patients who have a poor response to initial treatment for COVID-19; any treatment changes should be made after obtaining informed consent from the patient.”
Abstract
The conclusion from the abstract was LDH level on day 8 was the strongest predictor of in-hospital mortality in hospitalised COVID-19 patients, indicating its potential usefulness in post-treatment decision-making in severe COVID-19 cases. However, the conclusion from the manuscript says nothing about severe cases (We found that LDH levels on day 8 were the strongest predictor of in-hospital mortality 330 in hospitalized COVID-19 patients) since not every hospitalised patients has a severe disease. Moreover, in the methodology the severe form of COVID-19 is not defined and most patients included in this study had non-severe disease (81.8%, Table 1, variable Room air), at least at hospital admission as nothing else is mentioned about clinical form of the disease after this table and it is not included in the analysis of mortality which is also a major issue.
RESPONSE:
COVIREGI-JP is a nationwide registry system that has successfully collected information on many hospitalized COVID-19 patients in Japan. In reality, however, the circumstances of hospitalization of COVID-19 patients in Japan varied greatly depending on the region and the epidemic situation. At the beginning of the epidemic, all patients, even asymptomatic and mild cases, were hospitalized. Furthermore, the timing and threshold for hospitalization varied depending on the current pressure on the medical system. As a result, the status of registered hospitalized patients with COVID-19 is diverse, and the timing and severity of illness from the onset are also varied. The Japanese COVID-19 hospitalized patients were a heterogeneous group in terms of their status at the time of admission. Further, in some cases, patients with initially mild disease became severely ill after admission, making it difficult to accurately predict prognosis based on status and laboratory values at admission. In this study, the high mortality rate (6.8%) in the group with low LDH on day 1 and high LDH on day 8 may reflect that finding.
This study was conducted with the aim of finding a universal and simple prognostic marker that could be used to predict prognosis in a diverse group of hospitalized patients with varied statuses. In this context, we did not consider the severity of the disease; we incorporated only patient background and laboratory data for analysis and found that day 8 LDH alone had a high prognostic value. We believe that this concise study design is a strength of our study.
However, we included the results of respiratory status at admission in Table 1 to clarify the clinical picture of the patient groups.
Introduction section
The aim of the study is not well defined, in the abstract, but also in the manuscript.
RESPONSE:
Thank you for your observation. As you pointed out, the objectives were not stated in the Abstract between the background and methods of the study. We have revised the text to state that we examined the effect of laboratory trends on mortality. Also, we corrected the description in the Introduction to be provide better clarity.
Lines 18-19:
This study examined the laboratory values and trends that influence mortality in COVID-19 hospitalized patients.
Lines 73–75:
Using this registry data, our study aimed to validate two-time point laboratory values affecting mortality in hospitalized patients with COVID-19 and to find a universal and simple prognostic marker independent of broad patient status at admission.
Material and methods section
This section has a major issue for the analysis. The eligible patients were classified into four groups based on their LDH results on day 1 and day 8 (rows 107-108) and the authors mention different days for laboratory data collection Laboratory data on admission and day 8 included the results obtained on days 0–3 and 7–9, respectively (rows 91-92).
RESPONSE:
In COVIREGI-JP, data measured on days 0-3 are defined as day 1 and data measured on days 7-9 are defined as day 8, to allow for a reasonable margin in data collection. Day 0 refers to data from the day before admission, when a patient is transferred from another hospital. Since we were not able to ascertain exactly when the measurements were taken, we divided them into two categories, day 1 and day 8, for validation in this study. As you have pointed out, some of the descriptions in the Materials and Methods may create confusion therefore we have corrected them. Additionally, we have added this issue as a limitation in the Discussion.
Lines 97–101:
In accordance with the COVIREGI-JP data entry regulations, laboratory data collected on days 0–3 were pooled and registered as day 1 while those collected on days 7–9 were pooled and registered as day 8. Day 0 laboratory data refers to the data collected on the day before admission for a patient transferred from another hospital.
Lines 343–348:
Furthermore, as indicated in the Materials and Methods, there are discrepancies in the timing of the measurement of the laboratory values collected in this study. In addition, laboratory data other than day 1 (day 0–3) and day 8 (day 7–9) have not been validated; therefore, the possibility that there are other laboratory values and their timing that are optimal for prognostic prediction cannot be ruled out.
Results section
Other major issues could be found in Table 1:
- from the total of 8860 patients, 475 patients died (5.4% in hospital mortality rate) but only 35 patients were mechanically ventilated, assuming these patients were hospitalised in the ICU; what happened with the rest of the patients that died? Was that on a normal ward? (authors need to explain);
RESPONSE:
Most COVID-19 deaths in Japan are among the elderly. In Japan at that time, there were limited medical resources for ventilators to treat COVID-19, thus many elderly patients died without the use of ventilators. It is my understanding that many medical institutions treated COVID-19 patients in regular wards. There is an assumption that there were a significant number of cases of patients who would normally have had a ventilator indication but died without treatment. Given this notion that a large number of cases did not receive ventilators or other medical intervention even after deterioration, the outcome in this study was defined as in-hospital mortality. The actual situation is not clear from the registry database. This provides the motivation to find a universal and practical prognostic marker, despite not considering disease severity. However, as you have pointed out, the possible lack of appropriate medical intervention for severe disease should have been mentioned as a limitation, which was added in the Discussion.
Lines 348–350:
Last, in Japan, it is possible that a substantial number of severe disease cases that did not receive ventilators or other treatment were included. Therefore, the impact of LDH on mortality may have been overestimated.
- those are only descriptive values, a statistical test was not performed; the p values are missing; the authors could use a Kruskal Wallis test (?);
RESPONSE:
Thank you for your insightful feedback and suggestion. As mentioned above, in the current study we did not include severity (status on admission) in our multivariate analysis. Therefore, we did not consider it necessary to find statistically significant differences in severity of illness between the groups. Also, because of the large sample size in this study, even small differences are detected as statistically significant. In fact, almost all items were significantly different, and we did not consider listing p-values in the first draft. Following your suggestion, we have included them in Table 1.
Lines 124–125:
“Continuous and categorical variables were analysed using the one-way analysis of variance and Pearson’s chi-square, respectively.”
In the results section Considerable differences were observed in the median values of laboratory parameters (WBC count, lymphocyte fraction, platelet count, AST, ALT, LDH, CK, CRP, and creatinine) measured between the four groups on days 1 and 8 (Table 1) (rows 162-164). A statistical test was not used to support these results.
RESPONSE:
Thank you for your comment. As you pointed out, it was inappropriate to use the word “considerable” without showing a statistical difference. As noted above, p-values were not reported because all test values were judged to be statistically significant. We now have included them in Table 1.
Reviewer 2 Report
This is an interesting evaluation of factors related to COVID-19 associated mortality. The study is large, well-described and, although there are some limitations, the authors have appropriately defined them.
Minor comments:
Line 18. In a registry
Lines 69/284: Please explain "deviated enzymes" Perhaps abnormally elevated?
Lines 198/218. Use In contrast rather than Contrarily
Lines 235-8: Figure 3 legend is in the wrong place.
Author Response
Reviewer 2:
This is an interesting evaluation of factors related to COVID-19 associated mortality. The study is large, well-described and, although there are some limitations, the authors have appropriately defined them.
RESPONSE:
We are grateful to you for reviewing our paper, and will graciously accept any feedback and suggestions you may have to further improve our manuscript.
Minor comments:
Line 18. In a registry
RESPONSE: Thank you. We have revised the text accordingly.
Lines 19–20:
"Data on hospitalised patients enrolled in a registry study in Japan (COVID-19 REGISTRY JAPAN) were obtained.”
Lines 69/284: Please explain "deviated enzymes" Perhaps abnormally elevated?
RESPONSE: Thank you for your observation. We intended it describe an enzyme that leaked out of the cell, but have revised it to a more appropriate term.
Lines 49–51:
“In addition, patients with severe disease are more likely to have decreased lymphocyte counts and elevated levels of enzymes, inflammatory markers, and D-dimer."
Lines 299–300:
"LDH is a nonspecific enzyme, that is elevated in malignancies, liver diseases, interstitial pneumonia, and other infections.”
Lines 198/218. Use In contrast rather than Contrarily
RESPONSE: Thank you for the correction to a more appropriate wording.
Line 209–212:
“In contrast, when the prediction model was validated by employing laboratory values of both days 1 and 8, LDH levels on day 8 was selected as the variable with strongest effect on death among all the variables (adjusted OR [95% CI]: 1·006 [1·005–1·007], chi-squared value 250.747, p<0·001) (Table 2; appendix, Table S2).”
Lines 235-8: Figure 3 legend is in the wrong place.
RESPONSE: We have corrected it.
Reviewer 3 Report
I found this article to be a very interesting read and one that is certainly worthy of publication in Viruses. The data indicate that LDH is a useful predictor of prognosis of COVID-19.
I have a suggestion: in the Introduction section, the Authors talk about lymphocyte count and D-dimer (line 48-49). In my opinion, the Authors should add other important markers, such as suPAR, CRP and procalcitonin (Napolitano F, J Clin Med, 2021, 10(21), 4914).
Author Response
Reviewer 3:
I found this article to be a very interesting read and one that is certainly worthy of publication in Viruses. The data indicate that LDH is a useful predictor of prognosis of COVID-19.
I have a suggestion: in the Introduction section, the Authors talk about lymphocyte count and D-dimer (line 48-49). In my opinion, the Authors should add other important markers, such as suPAR, CRP and procalcitonin (Napolitano F, J Clin Med, 2021, 10(21), 4914).
RESPONSE:
Thank you for reviewing our paper, and for the insightful suggestion. We have cited and referenced the paper you suggested and stated that novel biomarker studies have also been reported.
Lines 51–52:
“Some previous studies have also focused on novel biomarkers such as soluble urokinase receptor and C-reactive protein isoforms.[10, 11]”